# *In Silico* Drug Design of Anti-Breast Cancer Agents

**DOI:** 10.3390/molecules28104175

**Published:** 2023-05-18

**Authors:** Kalirajan Rajagopal, Anandarajagopal Kalusalingam, Anubhav Raj Bharathidasan, Aadarsh Sivaprakash, Krutheesh Shanmugam, Monall Sundaramoorthy, Gowramma Byran

**Affiliations:** 1Department of Pharmaceutical Chemistry, JSS College of Pharmacy, JSS Academy of Higher Education & Research, The Nilgiris, Ooty 643001, Tamilnadu, India; 2Centre of Excellence for Pharmaceutical Sciences, School of Pharmacy, KPJ Healthcare University College, Nilai 71800, Negeri Sembilan, Malaysia

**Keywords:** breast cancer, benzothiophene analog, docking studies, pharmacophore modeling, 3D-QSAR, molecular dynamics

## Abstract

Cancer is a condition marked by abnormal cell proliferation that has the potential to invade or indicate other health issues. Human beings are affected by more than 100 different types of cancer. Some cancer promotes rapid cell proliferation, whereas others cause cells to divide and develop more slowly. Some cancers, such as leukemia, produce visible tumors, while others, such as breast cancer, do not. In this work, *in silico* investigations were carried out to investigate the binding mechanisms of four major analogs, which are marine sesquiterpene, sesquiterpene lactone, heteroaromatic chalcones, and benzothiophene against the target estrogen receptor-α for targeting breast cancer using Schrödinger suite 2021-4. The Glide module handled the molecular docking experiments, the QikProp module handled the ADMET screening, and the Prime MM-GB/SA module determined the binding energy of the ligands. The benzothiophene analog BT_ER_15f (G-score −15.922 Kcal/mol) showed the best binding activity against the target protein estrogen receptor-α when compared with the standard drug tamoxifen which has a docking score of −13.560 Kcal/mol. TRP383 (tryptophan) has the highest interaction time with the ligand, and hence it could act for a long time. Based on *in silico* investigations, the benzothiophene analog BT_ER_15f significantly binds with the active site of the target protein estrogen receptor-α. Similar to the outcomes of molecular docking, the target and ligand complex interaction motif established a high affinity of lead candidates in a dynamic system. This study shows that estrogen receptor-α targets inhibitors with better potential and low toxicity when compared to the existing market drugs, which can be made from a benzothiophene derivative. It may result in considerable activity and be applied to more research on breast cancer.

## 1. Introduction

Breast cancer is defined as a malignant tumor that starts in the cells of the breast. The type of breast cancer is determined by which cells in the breast becomes cancerous [1]. There are numerous locations in the breast where breast cancer can begin. Breasts primarily consist of lobules, ducts, and connective tissue. The milk travels through the ducts, which are tubes, from the breast to the nipple [2]. Connective tissue, which is made up of fibrous and fatty tissue, holds everything together. Usually, ducts or lobules are places where breast cancer develops [3]. Blood and lymphatic vessels are two ways that breast cancer can spread to other body areas. Metastasis refers to the spread of breast cancer to other bodily regions [4]. Cancer is a disorder wherein some body cells proliferate out of control and spread to other body regions [5]. In any one of the billions of cells that make up the human body, cancer can start almost anywhere. With more than 10 million deaths from cancer in the previous year, it is the leading cause of mortality worldwide. In all regions of India, the incidence of breast cancer has been rising by 0.5% to 2% annually in all age groups, but it has been especially high among women over the age of 45 years [6]. In the US, it is the second most common cause of death. By the end of the next five years, cancer incidences in India are expected to increase by 12%, according to the Indian Council of Medical Research (ICMR) [7]. Almost 23% of mortality in cancer patients is due to breast cancer. Many signaling mechanisms, including estrogen receptors (ER-alpha) and HER2 signaling pathways, which regulate stem cell proliferation, cell death, cell differentiation, and cell motility, control the normal breast development and mammary stem cells. ER-α (encoded by ESR1) is a crucial driver in oncogenic proliferation and metastasis, and about 70% of these individuals display it. The estrogen receptor, a nuclear hormone receptor, is divided into two types: estrogen receptor alpha (ER-alpha) and estrogen receptor beta (ER-beta). The estrogen receptor is involved in the development and maintenance of the female reproductive system [8], whereas ER- is mostly expressed in the prostate, bladder, ovary, colon, adipose tissue, and immune system. ER- is found in the mammary gland, uterus, ovary, bone, male reproductive organs (testis, prostate), liver, and adipose tissues [9]. Endocrine treatments, such as tamoxifen (TAM), have long been used to treat breast cancer by blocking estrogen binding to the receptor or preventing estrogen synthesis under aromatase catalysis [10,11]. Selected estrogen receptor degrader (SERD), such as fulvestrant, was discovered as a result of efforts to develop novel ER-α antagonist without this risk [12,13]. Tamoxifen, a selective estrogen receptor modulator (SERM), inhibits the E2-mediated activity of AF2, causing it to become ER-antagonistic while still retaining some partial agonistic effect. Contrary to tamoxifen, fulvestrant induces a change in the ER’s structure that interferes with both the transcriptional activity associated with AF2 and AF1 genes [14].

The ER-alpha receptor has lately received a lot of interest as a possible anti-cancer drug. The nuclear transcription factors estrogen receptors alpha (ER-alpha) and beta (ER-beta) are involved in the control of many complicated physiological processes in humans. The estrogen receptor subtypes alpha (ER-alpha) and beta (ER-beta) significantly influence the physiological effects of estrogenic substances. These proteins regulate the transcription of certain target genes in the cell nucleus by binding to related DNA regulatory regions [15]. Both receptor subtypes are expressed in many cells and tissues in humans, and they control key physiological functions in many organ systems, including the reproductive, skeletal, cardiovascular, and central nervous systems, as well as specific tissues (such as the breast and prostate, and ovary sub-compartments). The mammary gland, uterus, ovary (thecal cells), bone, male reproductive organs (testes and epididymis), prostate (stroma), liver, and adipose tissue are the primary sites of ER-alpha expression [16].

In this work, benzothiophene (BT) analogs, marine sesterterpene (MS) analogs, heteroaromatic chalcones (HC) analogs, and sesquiterpene lactone (SL) analogs have been discussed. These compounds were collected from literature studies that have inhibitory activities (IC50) in micromolar concentrations against breast cancer protein. These above-discussed analogs in our study target ER-alpha as their major target and possess inhibitory action. The majority of the compounds were far more effective against both drug-sensitive and drug-resistant breast cancer cells. The protein (PDB ID: 2IOG) was selected as the target as it possesses ER-alpha and was reported in the Protein Data Bank (PDB).

## 2. Results and Discussion

The results are summarized in Table 1, Table 2 and Table 3 and Figure 1, Figure 2, Figure 3, Figure 4, Figure 5, Figure 6, Figure 7, Figure 8, Figure 9, Figure 10, Figure 11, Figure 12, Figure 13 and Figure 14. The observation showed that the chemical makeup of the substituents had a significant impact on the compounds’ ability to inhibit breast cancer. The chemical structures of benzothiophene derivatives are given in Figure 1. The marine sesquiterpene analogs, heteroaromatic chalcones analogs, and sesquiterpene lactone analogs which have been tested are given in Figure 2, Figure 3 and Figure 4.

### 2.1. Molecular Docking Studies

For the purpose of assessing the compounds’ binding affinities at atomic levels, the ligands were docked to the active sites of proteins using the molecular docking program Glide module of Schrodinger suite 2021-4. To ascertain the inhibitory action of the developed analogs, they were docked to the breast cancer target (PDB ID: 2IOG). It is amply established that when compared to the standard drug tamoxifen, the compound BT_ER_15f has the highest Glide G-score (−16.14). BT_ER_15f represents BT-benzothiophene analog; ER is the target, and 15f is the compound code as given in Figure 1. The docking score and Glide G-score are given in Table 1 below which shows the best binding pose of the top 60 compounds. Figure 5 below represents the 2D and 3D docked poses of compound BT_ER_15f. The other 2D and 3D docked poses of the top 10 compounds are given in Appendix A.

**Table 1 molecules-28-04175-t001:** Molecular docking results for selected compounds against 2IOG.pdb.

S.No	Compound Code	Glide ENERGY	Docking Score	Glide-Gscore	XP H-Bond
1	BT_ER_15f	53.441	−15.922	−16.14	−1.546
2	BT_ER_Tf	59.574	−13.560	−13.563	−0.9
3	BT_ER_21b	65.999	−12.577	−12.964	−0.385
4	BT_ER_15e	51.353	−12.155	−12.825	−1.164
5	BT_ER_15b	57.5	−12.007	−12.776	−0.688
6	BT_ER_23c	44.431	−12.394	−12.404	−0.599
7	BT_ER_15d	60.801	−11.524	−12.321	−0.9
8	BT_ER_15c	52.788	−11.459	−12.32	−0.7
9	BT_ER_23b	49.559	−11.622	−11.632	−0.35
10	BT_ER_Rf	46.16	−8.852	−11.314	−0.627
11	BT_ER_21d	69.206	−10.469	−11.035	−0.335
12	SL_TN_55	27.163	−10.962	−10.962	0
13	SL_TN_56	27.163	−10.962	−10.962	0
14	SL_TN_34	15.699	−10.856	−10.856	0
15	BT_ER_23a	45.782	−10.773	−10.782	0
16	MS_ER_8b	18.796	−10.726	−10.726	0
17	SL_TN_51	26.449	−10.553	−10.553	0
18	SL_TN_63	26.449	−10.535	−10.535	0
19	BT_ER_21c	62.876	−9.931	−10.519	−0.605
20	BT_ER_15a	48.368	−7.952	−10.495	0
21	SL_TN_32	13.11	−10.492	−10.492	0
22	SL_TN_53	28.726	−10.462	−10.465	0
23	SL_TN_38	17.353	−10.447	−10.447	0
24	MS_ER_8a	24.947	−10.418	−10.418	−0.027
25	HC_TI_CT	52.763	−10.333	−10.333	−0.854
26	SL_TN_35	20.901	−10.255	−10.255	0
27	SL_TN_21	2.415	−10.214	−10.214	0
28	SL_TN_37	17.418	−10.146	−10.146	0
29	BT_ER_21e	61.13	−9.576	−10.101	−0.7
30	SL_TN_60	18.294	−10.065	−10.065	0
31	SL_TN_47	17.206	−10.043	−10.043	0
32	SL_TN_33_DETD_39	18.699	−10.04	−10.04	0
33	BT_ER_21a	56.951	−9.415	−10.027	−0.7
34	SL_TN_40	5.994	−9.892	−9.892	0
35	SL_TN_52	27.449	−9.861	−9.861	0
36	MS_ER_6a	24.224	−9.836	−9.836	0
37	SL_TN_39	24.888	−9.765	−9.765	0
38	SL_TN_59	35.939	−9.75	−9.75	0
39	SL_TN_57	29.053	−9.719	−9.719	0
40	SL_TN_58	29.053	−9.719	−9.719	0
41	BT_ER_25a	39.481	−7.193	−9.666	0
42	MS_ER_5b	19.258	−9.627	−9.627	0
43	MS_ER_4b	17.706	−9.603	−9.603	−0.178
44	SL_TN_44	12.471	−9.559	−9.559	0
45	SL_TN_42	10.596	−9.34	−9.34	0
46	SL_TN_31	11.769	−9.329	−9.329	0
47	SL_TN_41	8.182	−9.286	−9.286	0
48	SL_TN_27	5.437	−9.273	−9.273	0
49	SL_TN_16	9.358	−9.256	−9.256	0
50	SL_TN_19	4.803	−9.249	−9.249	0
51	SL_TN_12	4.923	−9.13	−9.13	0
52	SL_TN_46	8.311	−8.998	−8.998	0
53	SL_TN_20	3.842	−8.994	−8.994	0
54	BT_ER_25b	42.416	−8.96	−8.969	0
55	SL_TN_26	12.472	−8.959	−8.959	0
56	SL_TN_25	15.772	−8.941	−8.941	0
57	SL_TN_17	3.063	−8.894	−8.894	0
58	SL_TN_50	19.536	−8.723	−8.723	0
59	HC_TI_14	20.434	−8.713	−8.713	0
60	SL_TN_18	3.524	−8.71	−8.71	0

Glide energy; Docking score; Glide Gscore; XP H-bond (extra precision hydrogen bonding).

**Figure 5 molecules-28-04175-f005:**
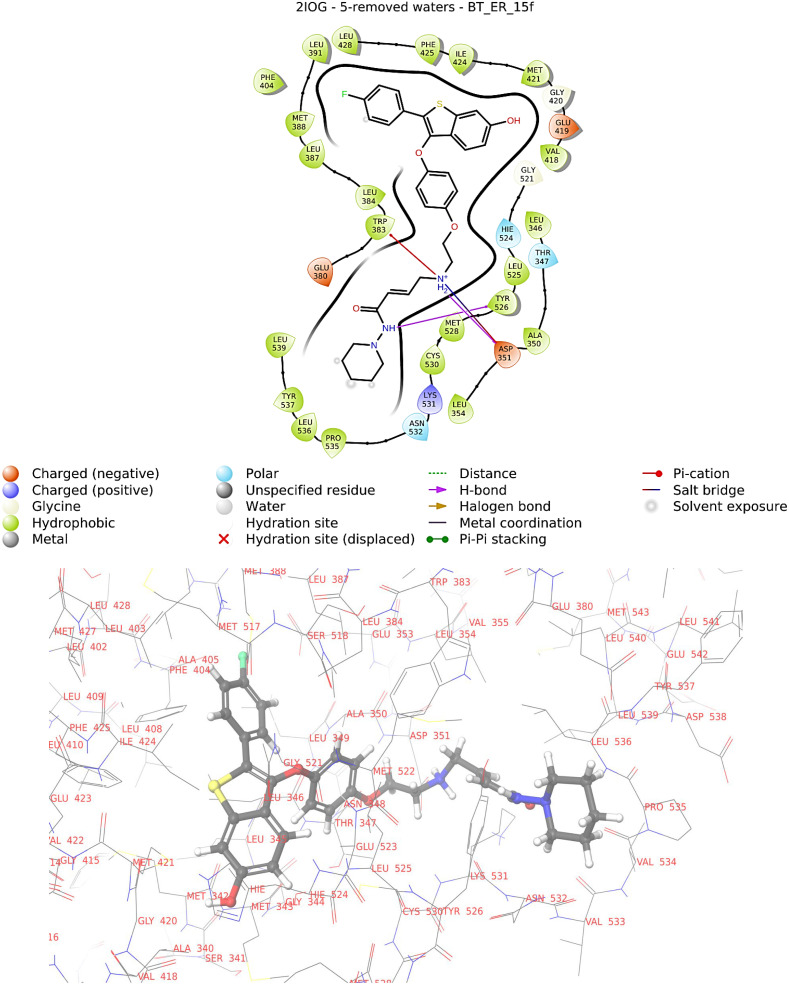
2D and 3D interaction diagram of BT_ER_15f with protein 2IOG.

The obtained Glide score is between −16.14 and −8.71, and the top score is for BT_ER_15f.

The amino acids residues binding LEU931, MET388, LEU387, LEU384, TRP383, LEU346, ALA350, LEU354, LEU530, TYR537, LEU536, PRO535, CYS530, MET528, TYR526, LEU525, VAL418, MET421, ILE424, PHE425, PHE404, and LEU428 make hydrophobic interaction with the ligand (BT_ER_15f). The amino acid residues THR347, ASN532, and (histidine) HIS524 make polar regions.

The lipophilic evidence of the aromatic moieties is what mostly causes the Glide scores to increase. The amino acid residues such as ASP351, GLU 380, and GLU419 form a negative charge around the ligand (BT_ER_15f_), and LYS 531 forms a positive charge around the ligand.

The discovered binding modes demonstrated that the ligand (BT_ER_15f) created connections with various residues LEU391 to LEU428 surrounding the active pocket through hydrogen bonds, hydrophobic interactions, and other mechanisms. The 2D and 3D interaction diagram of BT_ER_15f with protein 2IOG is given in Figure 5.

The amino group of the ligand (BT_ER_15f) binds to the active pocket with the amino acid residues TRP383 and ASP351.

### 2.2. Binding Free Energy Calculation Using MM/GBSA

Additionally, molecular docking was evaluated using MM/GBSA free restricting vitality, which is identified for breast cancer (PDB ID: 2IOG) target using a post-scoring approach, and the results are displayed in Table 2. The free energy of binding for the group of ligands was calculated using the Prime molecular mechanics-generalized Born surface area (MM/GBSA) of Schrödinger 2021-4 suite. The OPLS4 force field was used to minimize energy from the post-docked ligand-receptor complex with generalized-Born/surface area (MM/GBSA).
ΔG(bind) = E_complex_(minimized) − E_ligand_(minimized) + E_receptor_(minimized)

Because of the significant negative values produced by all compounds in the MM/GBSA experiment, the energies that showed strong ligand binding in the binding pocket of 2IOG are van der Waals energy (MMGBA dG Bind vdW) and non-polar solvation (MMGBA dG Bind Lipo). Other energies, such as covalent energy (MMGBA dG Bind Covalent) and electrostatic solvation (ΔGSolv), do not favor receptor binding. Moreover, greater negative values of MMGBA dG Bind vdW and MMGBA dG Bind Lipo demonstrate extraordinary hydrophobic interaction with 2IOG and ligands.

According to the findings of the MM/GBSA research, the DG bind values for considerably active compounds were found to be in the range of −15.33 to −84.12 kcal/mol. Additionally, dGvdW values, dG lipophilic values, and the energies are favorably contributing to the total binding energy [17]. BT_ER_15f, which has the highest docking score, exhibited excellent DG bind values of −70.59 kcal/mol.

**Table 2 molecules-28-04175-t002:** Binding free energy calculation using Prime MM-GBSA approach.

Compound	MMGBA dG Bind	MMGBSA dG Bind Coulomb	MMGBA dG Bind Covalent	MMGBA dG Bind H-bond	MMGBA dG Bind Lipo	MMGBA dG Bind vdW
BT_ER_15f	−70.59	−31.39	25.77	0.29	−47.46	−66.83
BT_ER_Tf	−73.77	−37.36	13.73	1.85	−50.23	−51.61
BT_ER_21b	−67.84	6.28	3.91	3.72	−44.3	−58.54
BT_ER_15e	−58.72	−7.14	11.68	0.65	−43.31	−50.6
BT_ER_15b	−84.12	2.23	12.68	1.55	−49.81	−67.59
BT_ER_23c	−35.85	35.14	4.56	2.43	−38.46	−38.31
BT_ER_15d	−69.35	15.23	16.5	1.6	−50.67	−76.44
BT_ER_15c	−77.87	−8.52	4.85	−0.1	−43.55	−51.96
BT_ER_23b	−42.39	4.45	15.46	2.18	−35.55	−43.15
BT_ER_Rf	−39.6	9.74	5.84	2.32	−40.64	−54.61
BT_ER_21d	−83.6	−23.01	9.8	−0.26	−46.56	−61.19
SL_TN_55	−47.68	19.48	13.24	1.19	−37.05	−63.86
SL_TN_56	−47.68	19.48	13.24	1.19	−37.05	−63.86
SL_TN_34	−47.89	2.46	17.92	0.06	−34.68	−54.25
BT_ER_23a	−22.06	43.87	4.15	3.73	−33.83	−37.62
MS_ER_8b	−51.07	23.16	16.5	1.72	−38.3	−53.5
SL_TN_51	−23.06	19.21	16.72	0.71	−27.94	−44.25
SL_TN_63	−32.26	7.75	28.36	0.71	−33.29	−60.91
BT_ER_21c	−62.02	−11.89	10.67	−0.17	−41.19	−40.9
BT_ER_15a	−48.48	−28.74	20.87	0.87	−40.95	−69.26
SL_TN_32	−51.78	21.56	10.52	2.17	−37.46	−60.5
SL_TN_53	−26.15	26.63	11.07	1.24	−26.98	−42.96
SL_TN_38	−43.82	36.19	3.87	4.46	−32.63	−52.89
MS_ER_8a	−32.26	13.04	8.48	3.91	−35.7	−31.91
HC_TI_CT	−37.49	33.67	9.55	4.97	−35.4	−59.22
SL_TN_35	−70.61	8.94	17.37	0.14	−35.16	−71.95
SL_TN_21	−40.68	33.51	17.41	1.79	−33.22	−58.98
SL_TN_37	−44.62	10.16	16.17	2.79	−35.66	−54.67
BT_ER_21e	−77.57	−12.32	8.39	−0.4	−42.39	−57.26
SL_TN_60	−15.66	30.43	11.01	2.91	−22.07	−33.97
SL_TN_47	−14.45	36.34	14.47	1.62	−26.13	−42.54
SL_TN_33	−44.77	−4.51	23.03	1.05	−37.47	−54
BT_ER_21a	−74.17	−25.99	19.61	0.51	−42.98	−64.65
SL_TN_40	−46.78	3.72	14.69	1.93	−34.26	−45.35
SL_TN_52	−61.66	51.17	6.73	2	−36.49	−64.87
MS_ER_6a	−80.51	45.68	7.44	3.79	−42.18	−69.73
SL_TN_39	−32.17	27.4	5.89	4.14	−29.4	−38.86
SL_TN_59	−23.46	16.16	−4.63	2.44	−23.08	−20.15
SL_TN_57	−45.01	24.7	13.93	3.5	−38.92	−47.17
SL_TN_58	−45.01	24.7	13.93	3.5	−38.92	−47.17
BT_ER_25a	−20.73	20.05	8.4	1.86	−32.66	−40.55
MS_ER_5b	−44.69	1.97	27.56	0.39	−37.99	−43.9
MS_ER_4b	−45.14	−4.23	26.93	−0.77	−36.71	−43.04
SL_TN_44	−51.74	29.57	3.53	1.71	−31.97	−53.17
SL_TN_42	−25.68	7.34	14.67	1.59	−25.3	−45.84
SL_TN_31	−59.09	26.08	34.57	0.18	−44.44	−62.36
SL_TN_41	−32.35	27.42	8.56	4.49	−31.66	−39.33
SL_TN_27	−36.14	19.28	8.97	3.86	−31.29	−44.2
SL_TN_16	−26.89	32.38	15.44	3.74	−32.99	−45.44
SL_TN_19	−28.82	41.14	10.29	3.07	−32.82	−41.2
SL_TN_12	−36.46	16.03	3.56	3.35	−29.29	−42.28
SL_TN_46	−17.05	31.24	14	1.77	−25.1	−43.97
SL_TN_20	−45.87	33.37	6.75	4.13	−31.83	−52.63
BT_ER_25b	−63.44	−23.53	13.99	0.3	−43.08	−57.07
SL_TN_26	−31.13	−2.41	17.3	0.95	−28.38	−56.03
SL_TN_25	−20.67	24.88	9.38	1.91	−21.97	−37.14
SL_TN_17	−24.26	41.87	−1.61	4.43	−25.99	−34.99
SL_TN_50	−40.32	3.61	8.68	1.26	−29.99	−35.15
HC_TI_14	−36.29	47.93	−4.62	5.56	−29.23	−33.83
SL_TN_18	−15.33	45.87	3.18	4.13	−25.08	−29.85

MMGBA dG Bind (free energy of binding); MMGBSA dG Bind Coulomb (Coulomb energy); MMGBA dG Bind Covalent (covalent energy); MMGBA dG Bind H-bond (hydrogen bonding energy); MMGBA dG Bind Lipo (hydrophobic energy); MMGBA dG Bind vdW (van der Waals energy).

### 2.3. ADMET Studies

ADMET features were predicted using the Schrödinger suite 2021-4’s Qikprop module. Properties such as molecular weight, dipole, hydrogen bond donor, hydrogen bond acceptor, log P o/w, and Lipinski’s rule of five are identified and mentioned in Table 3 below.

**Table 3 molecules-28-04175-t003:** *In silico* ADMET screening results of top 60 molecules using Qikprop module.

Compound	Mol MW	Dipole	Donor HB	Accpt HB	QP Log Po/w	Rule of Five
BT_ER_15f	561.67	5.605	3	7.5	5.995	2
BT_ER_Tf	355.522	0.865	0	2	6.682	1
BT_ER_21b	518.602	6.426	1	5	6.874	2
BT_ER_15e	563.642	4.866	3	9.2	5.013	2
BT_ER_15b	506.591	6.325	2	6.5	5.827	2
BT_ER_23c	463.522	7.699	2	4.5	6.319	1
BT_ER_15d	547.643	3.649	3	7.5	6	2
BT_ER_15c	533.616	4.877	3	6	6.059	2
BT_ER_23b	463.522	10.476	2	4.5	6.354	1
BT_ER_Rf	473.586	3.216	2	6.25	4.686	0
BT_ER_21d	559.654	6.407	2	6	7.16	2
SL_TN_55	488.536	6.502	0	8.75	3.567	0
SL_TN_56	488.536	6.502	0	8.75	3.567	0
SL_TN_34	490.432	8.923	0	8	3.939	0
BT_ER_23a	449.496	6.238	2	4.5	5.939	1
MS_ER_8b	454.648	4.201	0	6	5.814	1
SL_TN_51	444.483	5.698	0	8	3.007	0
SL_TN_63	444.483	5.698	0	8	3.002	0
BT_ER_21c	545.627	8.062	2	4.5	7.754	2
BT_ER_15a	492.564	7.054	3	6	5.127	1
SL_TN_32	440.879	6.971	0	8	3.291	0
SL_TN_53	445.471	7.567	0	9	2.305	0
SL_TN_38	450.487	7.079	0	8.75	3.325	0
MS_ER_8a	480.686	7.171	0	6	6.628	1
HC_TI_CT	348.357	9.984	1	7.75	1.742	0
SL_TN_35	472.46	6.031	0	6.75	4.818	0
SL_TN_21	426.508	7.617	0	8	3.504	0
SL_TN_37	450.487	6.55	0	8.75	3.323	0
BT_ER_21e	575.653	5.019	2	7.7	6.795	2
SL_TN_60	450.505	6.824	0	8	2.934	0
SL_TN_47	420.418	6.224	0	8.5	2.307	0
SL_TN_33	436.46	6.732	0	8.75	2.848	0
BT_ER_21a	504.575	7.879	2	4.5	6.882	2
SL_TN_40	396.396	7.295	0	8.5	2.042	0
SL_TN_52	474.509	5.824	0	8.75	3.042	0
MS_ER_6a	452.633	6.505	0	6	5.499	1
SL_TN_39	450.444	7.35	0	9.5	2.209	0
SL_TN_59	488.536	6.657	0	8.75	3.815	0
SL_TN_57	474.509	6.071	0	9.7	2.768	0
SL_TN_58	474.509	6.071	0	9.7	2.768	0
BT_ER_25a	461.507	7.977	1	4	6.11	1
MS_ER_5b	412.568	8.87	0	6	4.603	0
MS_ER_4b	398.541	3.782	0	6	4.268	0
SL_TN_44	410.423	6.063	0	8.75	2.101	0
SL_TN_42	386.419	6.579	0	8	1.975	0
SL_TN_31	420.461	7.979	0	8	3.112	0
SL_TN_41	412.456	7.171	0	8	2.707	0
SL_TN_27	386.444	6.974	0	8	2.236	0
SL_TN_16	386.444	8.057	0	8	2.481	0
SL_TN_19	402.486	6.561	0	8	2.848	0
SL_TN_12	358.39	8.016	0	8	1.783	0
SL_TN_46	459.292	6.371	0	8	2.614	0
SL_TN_20	388.46	6.587	0	8	2.382	0
BT_ER_25b	475.533	7.097	1	4	6.266	1
SL_TN_26	372.417	6.793	0	8	1.878	0
SL_TN_25	344.363	6.336	0	8	1.196	0
SL_TN_17	374.433	6.114	0	8	1.856	0
SL_TN_50	424.449	5.364	0	8.75	2.215	0
HC_TI_14	302.405	8.12	0	3.25	4.681	0
SL_TN_18	374.433	6.621	0	8	2.159	0

Mol MW (molecular weight of the molecule); Dipole (computed dipole moment); Donor HB (estimated number of hydrogen bonds that would be donated by the solute to water molecules in an aqueous solution); Accpt HB (estimated number of hydrogen bonds that would be accepted by the solute from water molecules in an aqueous solution); QP Log Po/w (predicted octanol/water partition coefficient); Rule of Five (Rule of Five Number of violations of Lipinski’s rule of five).

According to Lipinski’s rule of five, the molecule’s molecular weight should be ≤500, the partition coefficient should be ≤5, and the number of hydrogen bond donors and acceptors should be ≤5 and ≤10, respectively. All of these qualities, together with molecular flexibility, are thought to be important drivers of oral bioavailability. The BT_ER_15f ligand possesses a molecular weight of 561.67, a dipole moment of 5.605, an estimated number of hydrogen bonds that would be donated by the solute to water molecules in an aqueous solution is 3, and an estimated number of hydrogen bonds that would be accepted by the solute from water molecules in an aqueous solution is 7.5. With fewer exceptions, the obtained ADMET attributes are within the suggested ranges.

The number of H-bond donors is in the range of 0–2; the number of H-bond acceptors is in the range of 2–9.7. The number of violations of Lipinski’s rule of five is 0–2.

### 2.4. Pharmacophore Modeling

A pharmacophore model is a theory that explains how a group of compounds that bind to the same biological target exhibit the biological behaviors that have been observed [18]. The electron-withdrawing group, hydrogen bond donor, and hydrophobic top-active compounds are given below in Figure 6. The pharmacophore models were created using the Phase module of the Schrödinger suite 2021-4. The default set of six chemical properties of Phase was used to build pharmacophore sites for these compounds: hydrogen bond acceptor (A), hydrogen bond donor (D), hydrophobic (H) negative ionizable (N), positive ionizable (P), and aromatic ring (R). The distance and angles between different AAHHH.3 sites are shown in Figure 7a,b. AAHHH.3 represents that two hydrogen bond acceptors and three hydrophobic groups are essential for the activity. All ligands had their fitness scores evaluated using the AAHHH.3 model. A scatter plot analysis was also used to uncover discrete vital pharmacophoric requirements at spatial structure areas. The blue cubes around the ligand represented a favorable position for group substitution, whereas the red cubes showed a non-favorable position in Figure 6a–c for the top four ligands of this study.

**Figure 6 molecules-28-04175-f006:**
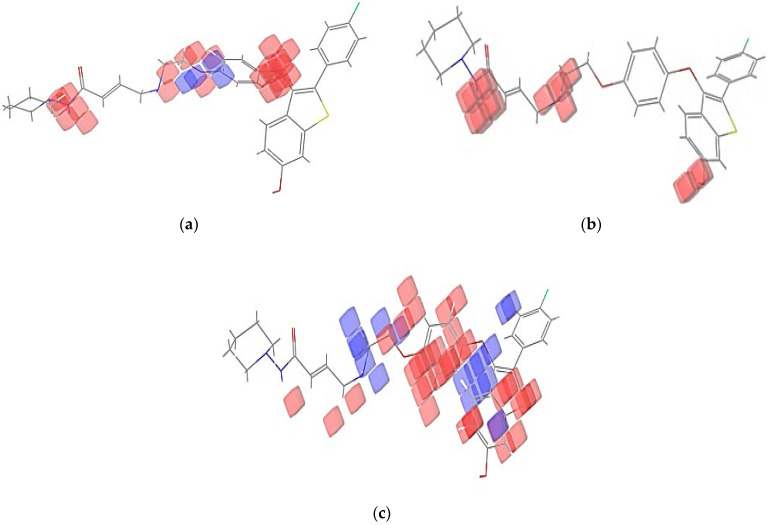
(**a**) Electron-withdrawing group favorable positions (blue colour); (**b**) Hydrogen bond donor group non-favorable positions for BT_ER_15f(red colour); (**c**) Hydrophobic group favorable (blue) and non-favorable (red) positions for BT_ER_15f.

**Figure 7 molecules-28-04175-f007:**
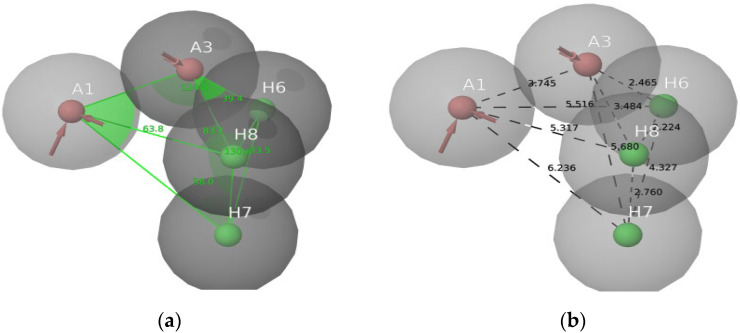
(**a**) Bond angle between the AAHHH.3; (**b**) Bond distance between the AAHHH.3.

### 2.5. 3D-QSAR Results

The atom-based QSAR module of Schrodinger suite 2021-4 was used to create the 3D-QSAR models for ER-alpha. Pharmacophore-based alignment of the ligands was taken into consideration in order to produce a statistically meaningful and highly predictive 3D-QSAR model [19]. Both the training and test sets of molecules had their prediction ability examined. Additionally, the default settings were applied, and a maximum of 2000 conformers and 15 conformations per rotatable bond were produced. Using vector, volume, site, survival, and survival in actives scores, the generated hypotheses were graded and ranked. Five places were determined to be common to all compounds in the hypothesis. A 3D-QSAR model was then developed using partial least squares (PLS) regression statistics.

The formula for the test set:y = 0.58x + 2.29 (R^2^ = 0.83)(1)

The green dots in Figure 8a,b represent ligands of the test set and training set. The ligands must be near the linear progression curve. The scatter plot with the XY-axis of the actual correlation with the predicted pIC50 is represented in Figure 8a,b for the test and training set compounds.

**Figure 8 molecules-28-04175-f008:**
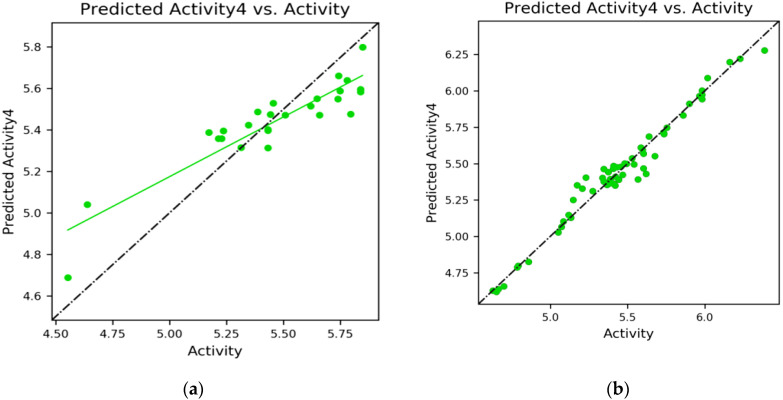
(**a**) Scatter plot for the test set; (**b**) Scatter plot for the training set.

### 2.6. MD Simulation

The MD simulation is used to estimate macromolecule mechanics, and it is based on classical mechanics and the application of Newton’s equation of motion to compute the speed and location of each atom in the investigated system. As a result, MD undertakes a more thorough structural investigation than docking, resulting in a more realistic depiction of protein motion [20].

Using a 100 ns MD, the stability of the docked BT ER 15f/2IOG complex was examined. Using the Desmond module of Schrödinger 2021-4, the complex in the explicit solvent system with the OPLS4 force field was investigated. The BT_ER_15f compound interacts with the protein residues as shown in Figure 9. The interaction fraction of each amino acid is given in Figure 10.

**Figure 9 molecules-28-04175-f009:**
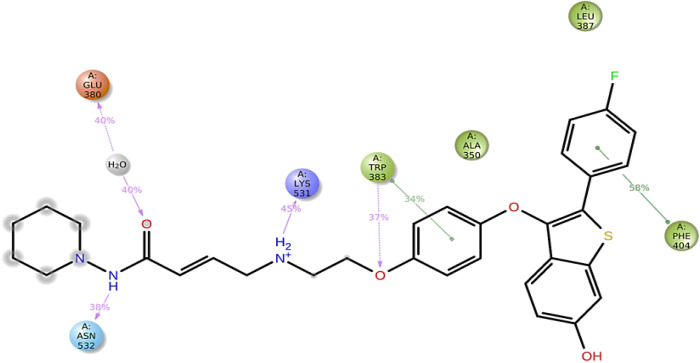
Ligand atom interactions with the protein residues.

**Figure 10 molecules-28-04175-f010:**
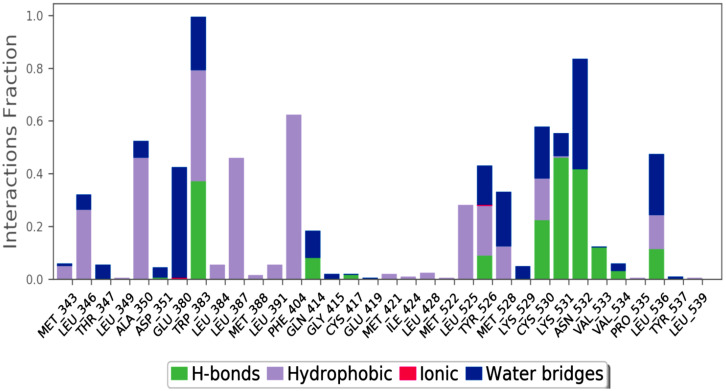
Interaction fraction of amino acids of BT_ER_15f.

The amino acid LYS 531, which is depicted in green in Figure 10, has the highest H-bond and a maximum interaction fraction of 0.5. H-bonds are essential for ligand binding. The donor and acceptor atoms in the donor-acceptor-hydrogen bond (D—H•••A) must be separated by 2.5 Å, the donor-acceptor-hydrogen bond (D—H•••A) must have a donor angle of 120°, and the hydrogen-acceptor-bonded atoms in the acceptor bond (H•••A—X) must have a donor angle of 90°. The following are the geometric requirements for hydrophobic interactions: p-cation, -aromatic, and charged groups that are within 4.5; p-p: two aromatic groups that are stacked face-to-face or face-to-edge; other non-specific hydrophobic side chains that are within 3.6 Å of a ligands’ aromatic or aliphatic carbons. A distance of 2.8 Å between the donor and acceptor atoms (D—H•••A), a donor angle of 110° between the donor-hydrogen-acceptor atoms (D—H•••A), and an acceptor angle of 90° between the hydrogen-acceptor-bonded atoms (H•••A—X) are needed for a H-bond to exist between a protein and water or water-ligand.

MD of standard tamoxifen was also performed, and it was found that the amino acids ALA350 and PHE404 have the highest interaction time. It is represented in Figure 11 and Figure 12. Amino acid residue ALA350 has a continuous interaction time. The RMSD value from the resulting trajectory analysis was in the range of 1.0 to 3.0. Green vertical bars in Figure 13 indicate protein residues that interact with the ligand, and the interactions between residues 100 and 130 showed the largest changes up to 2.4 Å. Through the formation of hydrophobic contacts with TRP383, ALA 350, PHE 404, and LEU 387, the molecule was positioned in the active pocket.

**Figure 11 molecules-28-04175-f011:**
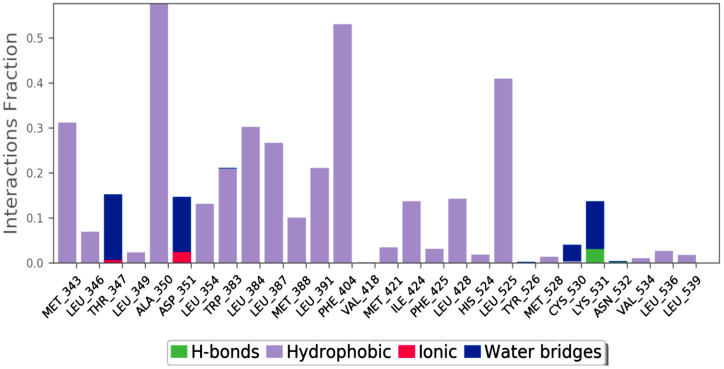
Interaction fraction of amino acids of the standard tamoxifen drug.

**Figure 12 molecules-28-04175-f012:**
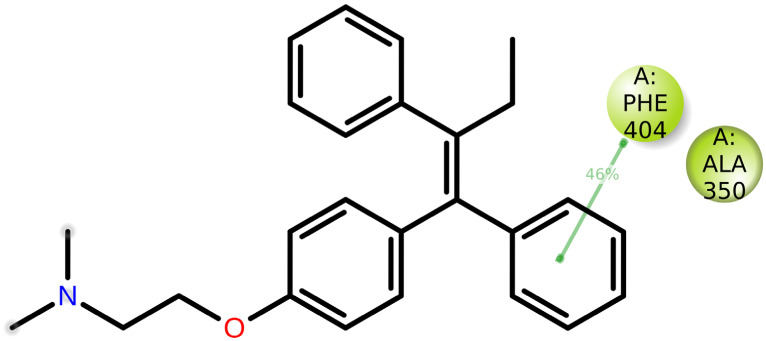
Standard tamoxifen interactions with the protein residues.

**Figure 13 molecules-28-04175-f013:**
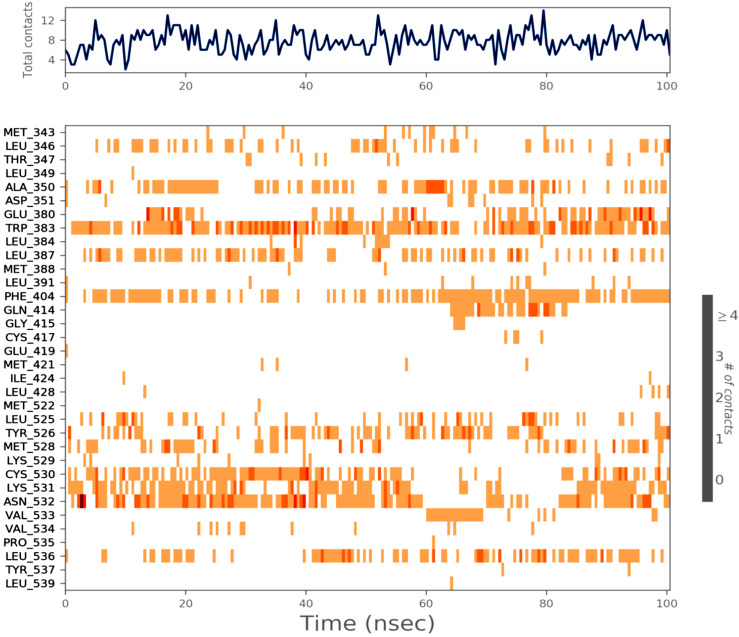
Interaction time of each amino acid.

At 37 ns, higher ligand RMSD fluctuations (up to 2.7 Å) were noted, given in Figure 14. Stable hydrophobic interactions with ALA 350, LEU 387, and PHE 404 were noted during the simulation. Utilizing measures of root-mean-square fluctuations, the flexibility of residues on ligand bindings was examined. To comprehend the molecular insights involved in the binding of TRP383 in the active pocket of protein target 2IOG, a 100 ns molecular dynamic simulation was conducted. 

It could be noted from Figure 15 that the deviation in the displacement of atoms is larger compared to BT_ER_15f. Thus BT_ER_15f, which has the best fit in the binding pocket, is best compared to the market available drug tamoxifen.

**Figure 14 molecules-28-04175-f014:**
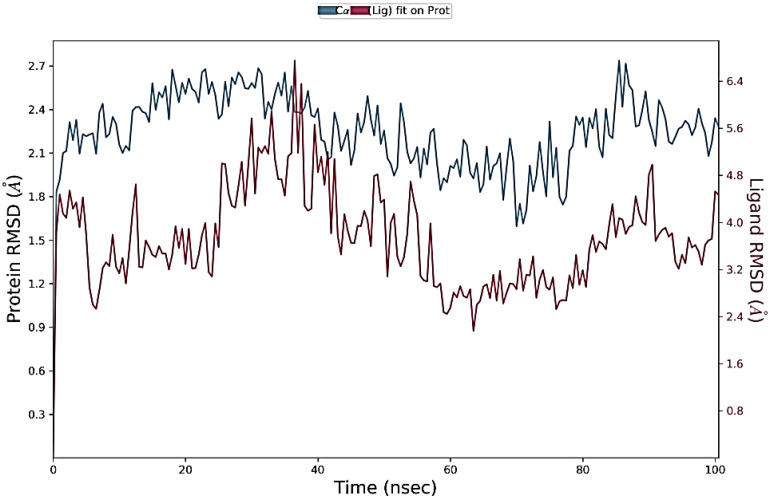
PL-RMSD of simulated protein 2IOG in complex with BT_ER_15f during 100 ns MD.

**Figure 15 molecules-28-04175-f015:**
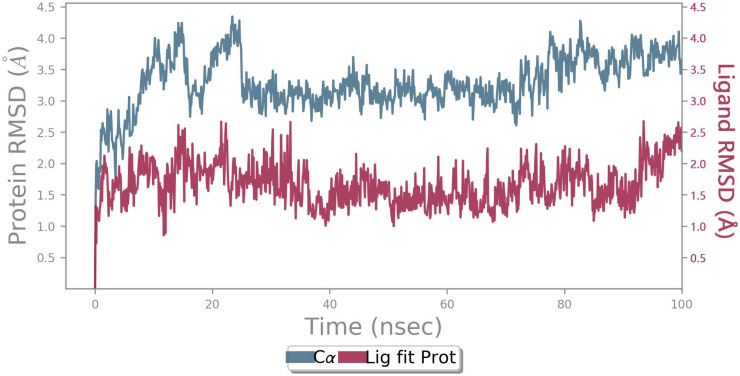
PL-RMSD of simulated protein 2IOG in complex with Tamoxifen during 100 ns MD.

The interaction time of each amino acid is given in Figure 13. It could be noted that the interaction times of amino acid TRP383 were greater than all other amino acids. Amino acid ASN 532 interaction was steady for the first 60 nanoseconds (ns), and then interaction was lost. Again, interaction occurs from 80 to 100 ns. Figure 16 provides the ligand (BT_ER_15f) characteristics such as ligand RMSD, radius of gyration (rGyr), intramolecular hydrogen bonds (intraHB), molecular surface area (MolSA), solvent accessible surface area (SASA), and polar surface area (PSA). The ligand and protein root-mean-square fluctuation is shown in Figure 17a,b, and it is important for describing local variations throughout the protein chain. RMSF is a measure of the displacement of a particular atom or group of atoms relative to the reference structure averaged over the number of atoms. RMSD is useful for the analysis of time-dependent motions of the structure.

L-RMSF (local root-mean-square fluctuation) and P-RMSF (protein root-mean-square Fluctuation) are both measures of a protein molecule’s flexibility or mobility. The average deviation or fluctuation in the position of each atom in a protein molecule from its average position in a given simulation or experimental data is measured as L-RMSF. It is calculated for a specific region or residue in a protein rather than the entire protein, and it is commonly used to identify flexible or disordered regions of a protein that are important for its function. P-RMSF, on the other hand, is the average RMSF value calculated for all the atoms in a protein molecule. It is used to quantify the protein’s overall flexibility or mobility and can aid in identifying regions that are relatively stable or flexible. P-RMSF can also be used to compare the flexibility of various proteins or conformations of the same protein. Both L-RMSF and P-RMSF are key techniques in the study of protein structure and function because they give insight into the dynamic features of proteins that are vital for their biological activity.

## 3. Materials and Method

### 3.1. Docking Studies

Docking studies were carried out mainly for four analogs which are marine sesquiterpene [21], sesquiterpene lactone analogs [22,23], heteroaromatic chalcones [24,25], and benzothiophene analogs [26] which were obtained from literature studies. The 3D crystal structure of the breast cancer protein 2IOG was previously co-crystallized with the *N*-[(1*R*)-3-(4hydroxyphenyl)-1-methylpropyl]-2-[2-phenyl-6-(2-piperidin-1-ylethoxy)-1*H*-indol-3-yl]acetamide. From the Protein Data Bank, the protein PDB ID 2IOG (resolution 1.6 Å) was retrieved. Arpita Roy published a paper on *in silico* investigation of agonists for proteins involved in breast cancer using the same target 2IOG [27]. The protein was optimized using the epic module of the Schrödinger suite 2021-4’s protein preparation wizard. By adjusting bond ordering, adding hydrogen atoms, and eliminating water molecules longer than 5 Å, the protein was optimized using the protein preparation wizard. Missing chains were then added using the Prime module of the Schrödinger suite 2021-4. The RMSD of the crystallographic heavy atoms was held at 0.30 for the OPLS4 molecular force field, which was used to minimize the protein. To pinpoint the centroid of the active site, a grid box was created. Using the Glide module of the Schrödinger suite 2021-4, all the compounds were docked into the catalytic pocket of the target protein 2IOG [28]. Significant Glide scores indicate ligands with higher 2IOG binding affinities [29,30]. Hussein highlighted in his work that the compound CH4 (chalcone) exhibited a binding energy of −10.83 kcal/mol against the target protein 2IOG [31], which possesses anti-breast cancer activity. 

### 3.2. MM/GBSA Binding Free Energy Calculation

The precise determination of binding free energy plays a very essential role among the several strategies that may be used to analyze the ligand-receptor interaction [32,33]. Using the Prime molecular mechanics-generalized Born surface area (MM/GBSA) of Schrödinger 2021-4, post-docking energy minimization calculations were carried out to determine the free energy of binding for the collection of ligands in complex with a receptor [34]. The Poisson–Boltzmann surface area (MM/PBSA) and molecular mechanics-generalized Born surface area (MM/GBSA) are arguably very popular methods for binding free energy predictions [35]. Imaobong Etti published that the Artocarpus species has good anti-cancerous properties [36]. He and his co-workers found that Artonin E possesses the best drug-likeness using the Prime module of the Schrodinger software 2021-4 against the target protein 2IOG.

### 3.3. In Silico Predicted ADMET Properties

By identifying the most promising candidates for development and eliminating those with a low chance of success, early assessment of ADME-Tox characteristics can reduce the time and expense of screening and testing. The regulatory authorities are now very interested in the practical application of *in silico* methodologies for predicting preclinical toxicological endpoints, clinical side effects, and ADME features of new chemical entities [37]. ChemAxon properties such as molecular weight, total polar surface area (TPSA), hydrogen bond acceptor and donor count, log P, log D, log S, molar volume, and dissociation constant (KD), as well as the number of violations of Lipinski’s rule of five, van der Waals volume, and other properties were used to predict the physically and pharmacokinetically significant descriptors for the top hits by using the Qikprop module of Schrodinger suite 2021-4. Table 3 displays these outcomes.

### 3.4. Pharmacophore Modeling

An explanation for the pharmacological effects of a collection of substances that bind to the same biological target is known as a pharmacophore model [38]. “An ensemble of steric and electronic features necessary to produce optimal supramolecular interactions with a given biological target” is a pharmacophore model [39]. A pharmacophore model can then be used to query the 3D chemical library to look for potential ligands, which are referred to as “pharmacophore-based virtual screening,” depending on whether the approach was ligand- or structure-based virtual screening (VS) [40]. The pharmacophore model was created using the Phase module of the Schrodinger suite 2021-4. The common pharmacophore AAHHH.3 found from our work can be used for further high-throughput screening to screen a large database [41]. Tien-Yi-Hou and his co-workers performed work on estrogen receptor-α ligand binding through pharmacophore modeling and concluded few pharmacophore models active against breast cancer.

### 3.5. QSAR-Quantitative Structure Activity Relationship

The application of force field calculations requiring three-dimensional structures of a given collection of small molecules with known activities is referred to as 3D-QSAR. 3D-QSAR is an extension of classical QSAR that uses robust statistical analysis such as PLS, G/PLS, and AN to explain the three-dimensional features of ligands and predict their biological activity [42].

A computational modeling method known as the quantitative structure-activity relationship (QSAR) helps researchers connect the structural characteristics of chemical compounds with their biological functions. Drug development requires QSAR modeling [43].

### 3.6. Molecular Dynamics

A technique for simulating the physical motions of atoms and molecules is called molecular dynamics (MD) [44]. For a predetermined period of time, the atoms and molecules are allowed to interact, giving insight into the dynamic “evolution” of the system. Molecular dynamics is a method for computing the time evolution of a group of interacting atoms using Newton’s equations of motion. In the thermodynamic process of protein-ligand interaction, a tiny molecule’s solvation free energy acts as a stand-in for the ligand’s desolvation [45]. MD of the highest Glide score compound was performed in this work using the Desmond module of Schrodinger suite 2021-4. From these geometric requirements around the ligand, BT_ER_15f was identified.

## 4. Conclusions

In the realm of drug design and discovery, integrated methodologies of QSAR and molecular docking-based prediction have been successfully used in a number of statistically supported examples. The current research on benzothiophene analogs, specifically BT_ER_15f, using molecular docking and QSAR demonstrated that it has a sizable anti-cancer effect against the target 2IOG.

From the docking study, the benzothiophene derivative demonstrated better arrangement at the dynamic site. The current investigation aided in identifying the key compounds and their beneficial effects. In subsequent analysis using in vitro and in vivo techniques, it could be optimized as a drug to treat breast cancer. According to the findings, the compound BT_ER_15f, a benzothiophene derivative, exhibits strong anti-breast cancer action and is useful for future research.

The pharmacokinetics and drug-likeness studies revealed that the ligand BT_ER_15f could be the best drug candidate against breast cancer.

In the future, this study will be a reliable resource for achieving further benzothiophene derivatives through innovative structural modifications in benzothiophene derivatives that are being widely researched. These findings provide compelling support for novel studies that involve developing more methodological frameworks to investigate molecular facets of their anti-cancer action.

## Figures and Tables

**Figure 1 molecules-28-04175-f001:**
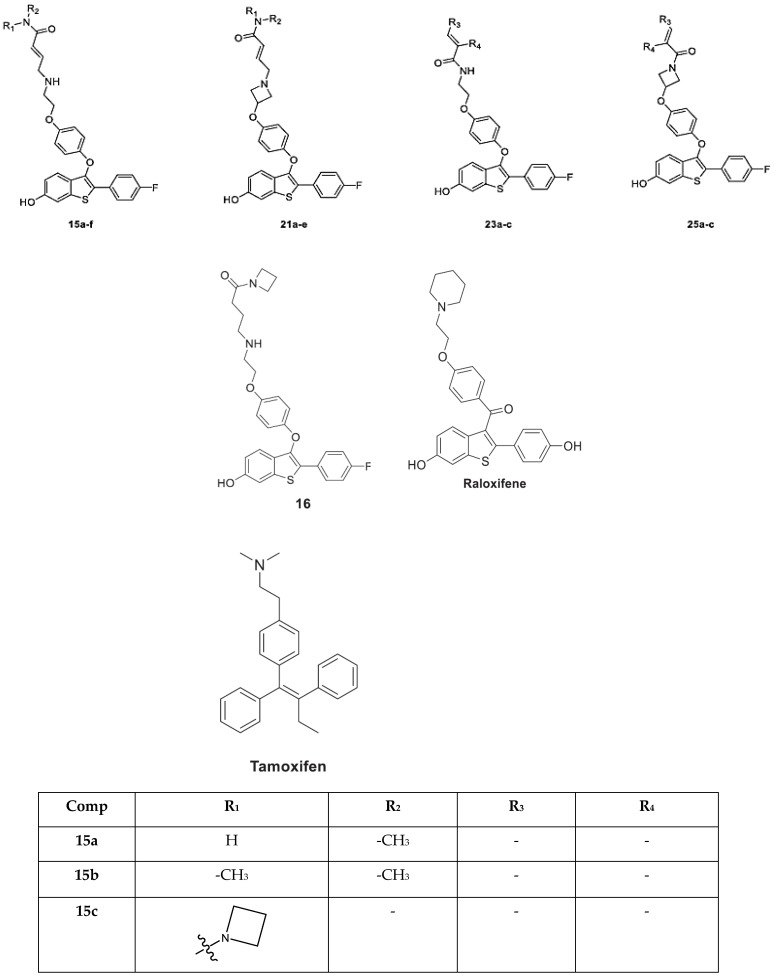
Structure of benzothiophene (**BT**) analogs.

**Figure 2 molecules-28-04175-f002:**
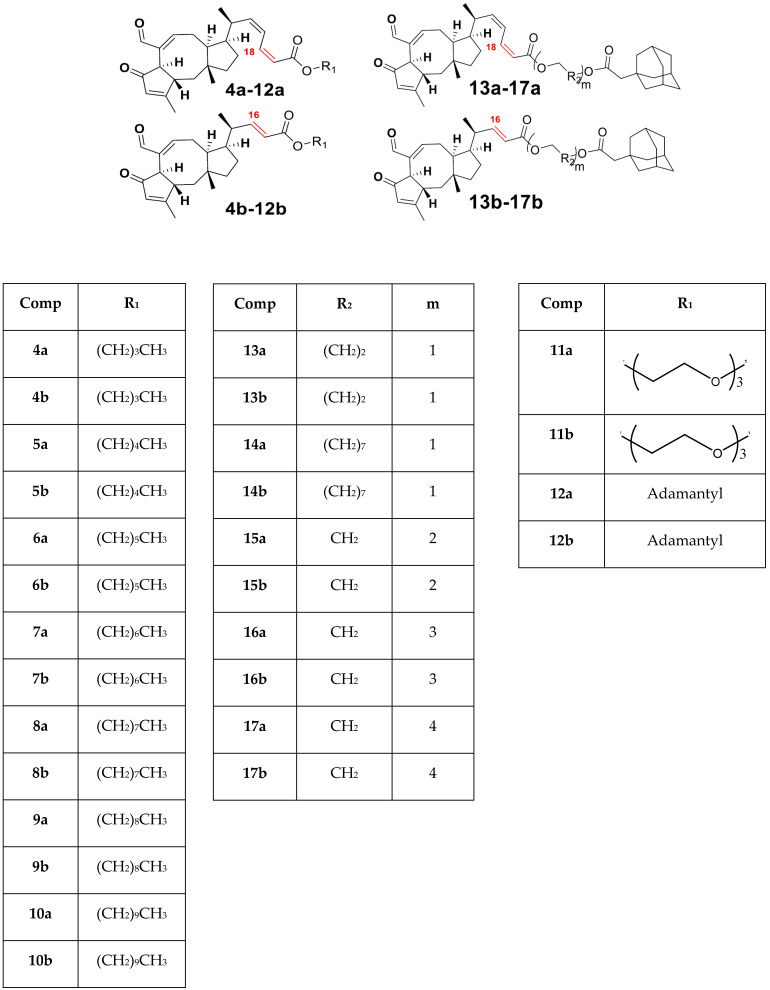
Structure of marine sesterterpene (MS) analogs.

**Figure 3 molecules-28-04175-f003:**
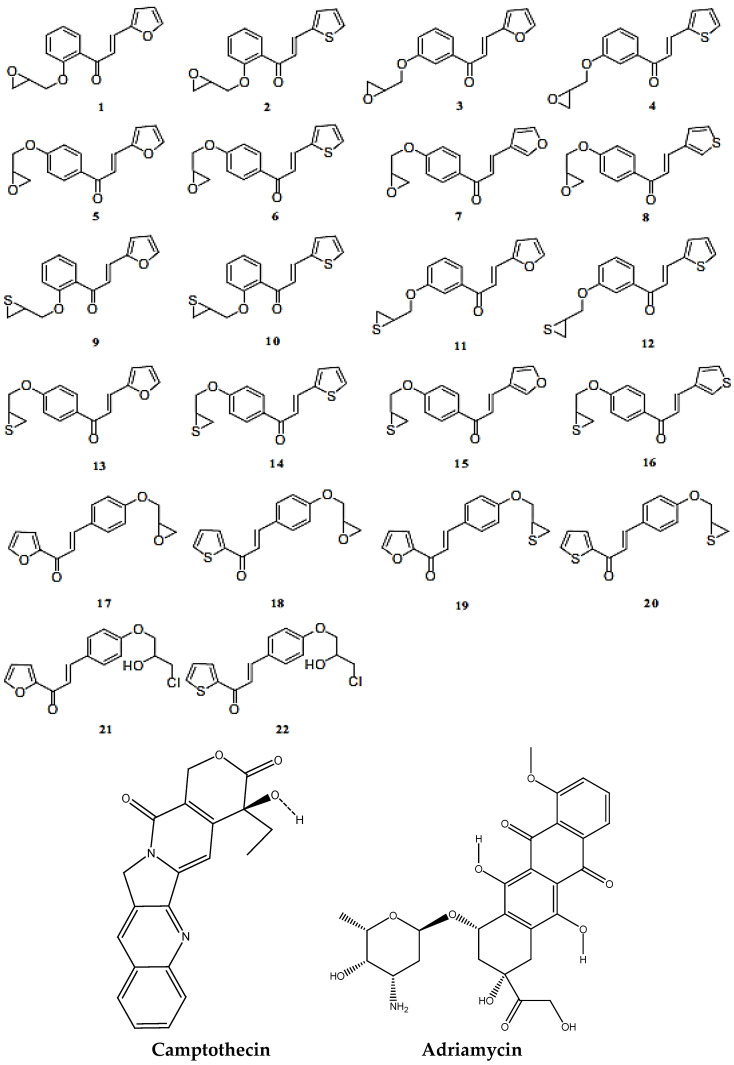
Structure of heteroaromatic chalcones (HC) analogs.

**Figure 4 molecules-28-04175-f004:**
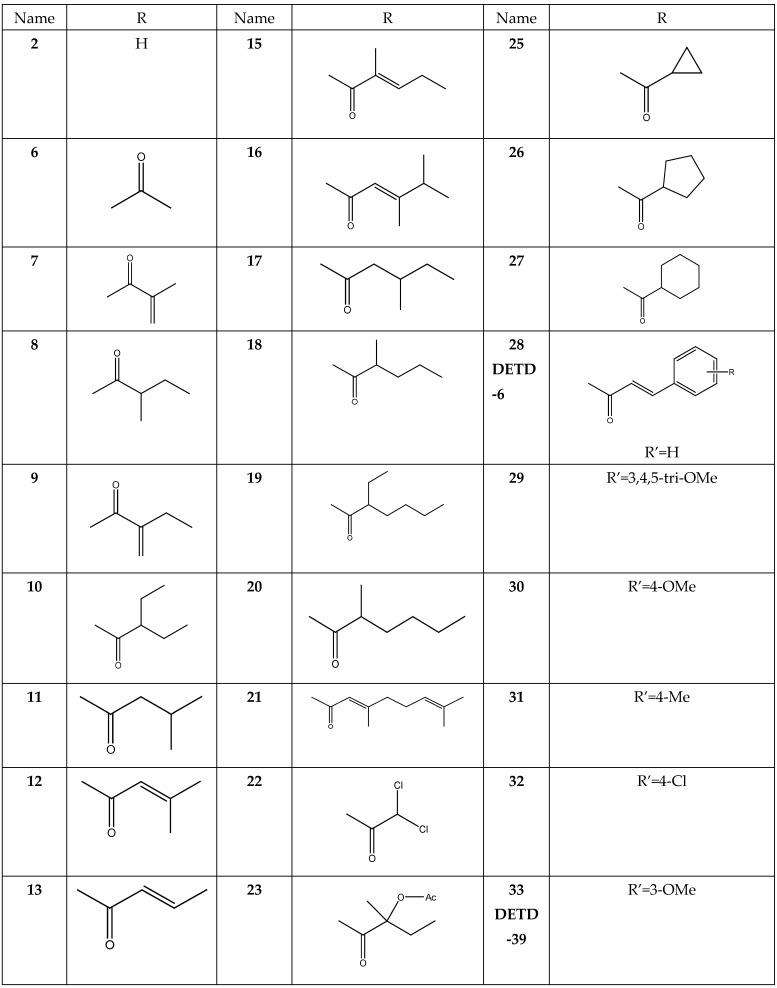
Structure of sesquiterpene lactone **(SL)** analogs.

**Figure 16 molecules-28-04175-f016:**
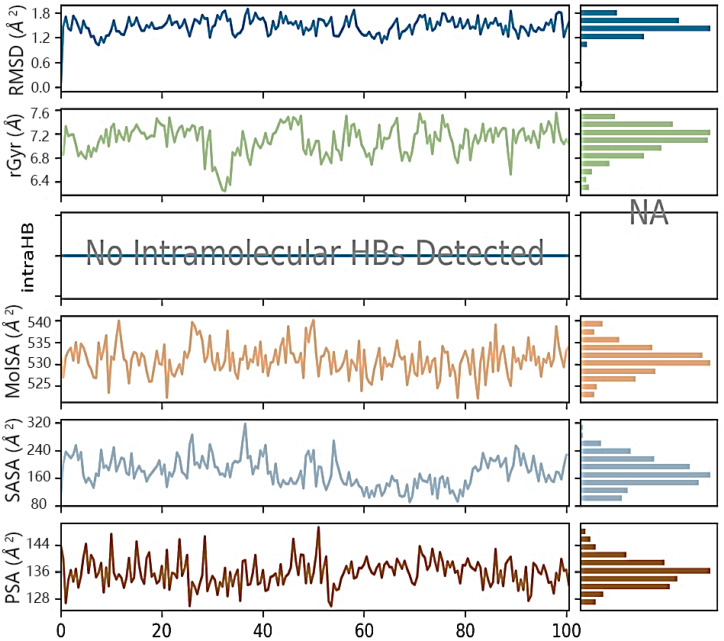
Ligand properties of BT_ER_15f in complex with 2IOG during 100 ns MD simulation.

**Figure 17 molecules-28-04175-f017:**
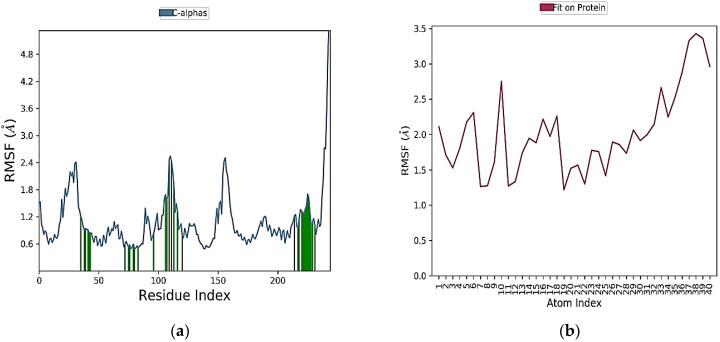
(**a**) L-RMSF of simulated protein; (**b**) P-RMSF of simulated protein 2IOG in complex with BT_ER_15f.

## Data Availability

The data supporting reported results can be available with corresponding author Kalirajan Rajagopal which will be shared on request by mail.

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
