# Peer review of "In Silico Drug Design of Anti-Breast Cancer Agents"

_molecules, 2023, doi:10.3390/molecules28104175_

Round 1
Reviewer 1 Report
1- Figure 1 and its compounds begin from 15a and it seems that the authors snap it from references and have to redraw it with a suitable program.
2- figure 2 are wholly unorganized and must be fixed appropriately.
3- figure 3 have to be fixed as I described before.
4- figure 4 has to be redrawn with any drawing program as described before.
5- The table 3 title has to change to describe the table content.
6- the conclusion is very concise and doesn't describe the result well and has to be changed.
Author Response
1- Figure 1 and its compounds begin from 15a and it seems that the authors snap it from references and have to redraw it with a suitable program.
Reply: Redrawn using chemdraw software and have made changes.
2- figure 2 are wholly unorganized and must be fixed appropriately.
Reply: Changes made in the manuscript as per reviewer’s comment
3- figure 3 have to be fixed as I described before.
Reply: Changes made in the manuscript as per reviewer’s comment
4- figure 4 has to be redrawn with any drawing program as described before.
Reply: Redrawn using chemdraw software.
5- The table 3 title has to change to describe the table content.
Reply: Changes made in the manuscript as per reviewer’s comment
6- the conclusion is very concise and doesn't describe the result well and has to be changed. Reply: The description of the result have been changed and highlighted.
Reviewer 2 Report
attached

Author Response
Abstract
- The abstract should not have references. The author should check journal guidelines.
Reply: Reference removed in the abstract as per reviewer’s comment
- Name the protein structure instead of PDB ID in the abstract
Reply: Changes made in the manuscript as per reviewer’s comment
- Provide unit for binding energies in the abstract
Reply: Docking score unit Kcal/mol. are included.
- In abstract, what do you mean by “tamoxifen (-13.560)
Reply:- -13.560 is Glide score of tamoxifen.
TRP383 has highest interaction time with the ligand hence it is more important amino acid residue” which amino acid? The sentence doesn’t make any sense –
Reply: Changes made in the manuscript as per reviewer’s comment (TRP means tryptophan)
Introduction
- In the introduction provide standard medical definition of breast cancer.
Reply: Changes made in the manuscript as per reviewer’s comment
- In introduction you said “Breast cancer comes in a variety of forms” later no forms are elaborated ?
Reply: Changes made in the manuscript as per reviewer’s comment
- “Which breast cells develop into cancer determines the type of breast cancer” correct grammatically.
Reply: Changes made in the manuscript as per reviewer’s comment
- I cannot see any reference in ¾ part of introduction.
Reply: References are included as per reviewer’s comment.
- Statistics of breast cancer have no reference.
Reply: References are included as per reviewer’s comment
- There is no introduction for ligands related to Marine sesquiterpene, sesquiterpene lactone, heteroaromatic chalconesand benzothiophene so as there is no studies referred –
Reply: Added in introduction section as per reviewer’s comment
- There is no introduction for receptor and why is it target, why is important, what we previously know about it
Reply: Added points in introduction section and highlighted as per reviewer’s comment
- In short Introduction needs to be rewritten completely
Reply: Rewrite the introduction as per reviewer’s comment
- In first figure with no caption what is 15a-f, 21 a-e and so on… there is no detail about them and in my impression these are just copy and pasted and same number is given as in some previous paper… Justify this figure here … if necessary draw it yourself and tell us reason why it is here –
Reply: Redrawn using chemdraw software and changes made accordingly. Those are the compounds selected for in-silico design which we are analyzed.
- Figure 1 look like a table to me for previous figure without any number. again why these number start with 15 ?
Reply: The numbering are given based on the literature collected. From the literature review, we are collected 100s of molecules and given only significantly active molecules by in-silico method. The figures given were the top molecules in each derivatives like Benzothiophene (BT) analogues, Marine Sesterterpene(MS) analogues, Heteroaromatic chalcones (HC) analogues, Sesquiterpene lactone (SL) analogues according to the results obtained from in-silico studies.
- What are bases for these analogues?
Reply: The details are Mentioned in the manuscript
- Figure 2 is also haphazard.. Should be revisited completely … no info is given about it
Reply: Redrawn using chemdraw software and changes made accordingly.
- Figure 3 .why authors are making some structure small and some big ..?some with colored functional groups some without Why numbering is inappropriate ?
Reply: Redrawn using chemdraw software and changes made accordingly.
- Figure 4 has same problems.
Reply: Redrawn using chemdraw software and changes made accordingly..
- These figures 1-4 are cited inappropriately in the text.
Reply: figures 1-4 are cited accordingly as per reviewer’s comment.
Methods
- Authors have never named 2IOG or given its importance in the paper .
Reply: name of 2IOG is given in introduction
- In start of methods .Authors claimed “Breast cancer protein 2IOG 3D crystal structure is co-crystallized with the N- [(1R)- 3-(4 HYDROXYPHENYL)-1-METHYLPROPYL]” I don’t think they really did it?? Explain –
Reply: It was already co- crystallized and is available in PDB(Protein data bank).
- What authors mean by “The protein was made using the epic module of the …” correct this ..DONE so as “and eliminating water molecules longer than 5”.
Reply: Changes made in the manuscript as per reviewer’s comment
- “the protein was created using the protein preparation wizard” how possibly you can create a protein ??? you need to correct yours statements
Reply: Changes made in the manuscript as per reviewer’s comment
- Methods with Schrödinger suite are not appropriately referred. References 16-17 are inappropriate here
Reply: Changes made in the manuscript as per reviewer’s comment
- Why MM/GBSA is written as MM-GB/SA.. its abbreviation as well as full version is not correct completely
Reply: Changes made in the manuscript as per reviewer’s comment
- No method given for pharmacophore modeling ?
Reply: The method is given in the results section.(3.4) Some points were included according to the reviewer comment.
- No proper method or reference is given for QSAR ?
Reply: References are added as per reviewer’s comment
- How MD were performed there is no method given in 2.6 ??
Reply: The method is given in the results section.(3.6) Some points were included according to the reviewer comment.
- In short, methods should be rewritten. –
Reply: Changes made in the manuscript as per reviewer’s comment
Results
- The first sentence of the results look like a conclusion “The findings showed that the chemical makeup of the substituents had a significant impact on the compounds' ability to inhibit breast cancer.
Reply: Changes made in the manuscript as per reviewer’s comment.
- The result figure numbers are not cited properly in text
Reply: figure numbers are cited in the manuscript as per reviewer’s comment
- I can’t see BT ER in derivative figure ? In results there is even no details about it – Reply: BT ER is representing the code such as BT- Benzothiophene analogues ER- Estrogen receptor which eas included in the manuscript.
- Figure on page 9 is without number?
Reply: Figure 5 is 2D and 3D interaction diagram of BT_ER_15f with protein 2IOG which was included in the manuscript
- I cant see anything in Figure 5 .. redraw it properly so it will show something
Reply: Figure 5 is improved to clear picture.
- Explain what is glide score and how it will relate to binding energy.
Reply: Glide score is the binding energy in Kcal/mol
- Figure resolution is very bad.. these are not readable.. I can only read them once these are revised
Reply: Figure resolution is improved as clear picture.
- What is HIE524residue ?CORRECTED in “The amino acid residues THR347, ASN532 and HIE524 makes polar region.” Show this polar region in figure and say why it is important .
Reply: It is given in the figure 5 that the blue colour aminoacids represents the polar groups.
- As I cant see a separated discussion section, author should have discussed there results somewhere.. but I cannot see any discussion … this part should be sparely added or author should discuss results here .
Reply:There isn’t a separate discussion section instead it is discussed under the each and every topic heading.
- “ ligands created connections with various residues LEU 391 to LEU 428” which ligands.. where and how ?
Reply: Some Hydrogen bonding interactions are formed between ligand and aminoacid ASP351 and TRP383. The residues LEU 391 to LEU 428 is referes as bindling pocket which is clearly shown in 2D interaction diagram of the figure 5.
- Where are docked poses for other ligands ?? what are their interactions . are the polar, non polar patches or charged interactions are common ?? explain with figures.. you may add figures in supplementary material
Reply: The 2D and 3D docked poses for other ligands (Top 10) with their interactions are given in figures 5a-5j in the supplementary material as per reviewr’s comment
- How MM-GB/SA was calculated without simulations? I cannot see method, or method details, references? This is important to mention all these.
Reply: It is given in MM-GB/SA section and highlighted.
- Can you compare MMGBSA values with previous studies. which value is a threshold and which value is least? which one is higher and why and what does it means ?
Reply: Explained and highlighted in the MM/GBSA section.
- There is no discussion for ADMET features, which is important and why and what are values for your potent ligand ? is it different from others and why?
Reply: Discussed and highlighted in ADMET studies.
- The compound codes in table 1 are not given else in any figure and have not been shown what are those ?
Reply: Those are the analogues chosen like Benzothiophene (BT) analogues, Marine Sesterterpene(MS) analogues, Heteroaromatic chalcones (HC) analogues, Sesquiterpene lactone (SL) analogues which was included in figures 1 to 3.
- Explain all column heading in the footnote of table 1-4
Reply: foot note is included below each table and highlighted.
- Some compound have lesser MMGBA dG than your mentioned ligand.. what are reasons ? is it means there are more stable ligand than the one you mentioned ?
Reply: Yes there are lesser MMGBA dG than our mentioned ligand which has good binding but it doesn’t satisfy the other requirements.
- Explain phagophore modeling figure .. how the final pharmacophore was developed .. please discuss. How many model were obtained .. what were parameters of optimization
Reply: Discussed above the figure 6a.
- What colors say in figure 6
Reply: Explained in last line above figure 6 of pharmacophore modeling.
- What is “AAHHH” in figure 7 ? explain
Reply: AAHHH is the best pharmacophore model with two aromatic rings and three hydrogen bond acceptors which id Explained and highlighted
- What is “3.5.3. D QSAR results” what is D?
Reply: It is section 3.5, 3D QSAR - correction made in the manuscript
- Explain figure 8 what does it says ?
Reply: Described in the manuscript
- Why simulations only for BT ER 15f/2IOG complex were performed ?
Reply: The MD simulations was performed for only top active compound.
- Explain/cite figure 9 and 10 in the text .. discuss them with respect to docking.
Reply: It is related to molecular dynamics and is cited and discussed below the figure 10.
- “At 37 ns, higher ligand RMSD fluctuations (up to 2.7 Å )” I cant see this where ? which figure ? can you please mention
Reply: It is given in figure 14 which is mentioned and highlighted.
- “The interactions between residues 100 to 130 showed the largest changes up to 2.4 Å” I cannot see these either?
Reply: It is given in figure 14 which is mentioned and highlighted.
- Give figure 11 after citation. Discuss it with previous studies? Why this is important Reply: It is described in the manuscript
- What do you means by “Figure 12 provides the Ligand characteristics.” Explain it please –
Reply: It is Explained and highlighted in the manuscript .
- Are RMSF show same results as RMSD ? why difference” any explanation ? discussion
Reply: RMSF- Root mean squire Fluctuation, RMSD- Root mean squire Deviation which is explained and highlighted.
- How P and L – RMSF are different?discuss
Reply: Explained and highlighted.
- I can not see any discussion in the result part. many things need to be revised thoroughly
Reply: The discussions are revised and highlighted
Reply to General comments
- The English language and style were improved as per reviewer’s comment
- Additional references were added as per reviewer’s comment
- The figures are improved as per reviewer’s comment
- Citations are corrected as per reviewer’s comment
- Additional descriptions are included in many sections as per reviewer’s comment.
Reviewer 3 Report
The article describes by means of in silico methodologies the binding model of some molecules to the 2IOG protein.
In this sense, it is considered that the manuscript should clarify the methodologies used, describing them in more detail.
In the results section, a greater description and discussion of the same is necessary, being supported with references to recent studies and works.
In the case of the 3D-QSAR results, activity results are used, which are not previously presented or cited in the manuscript before their presentation in the figure 8.
I consider it convenient to increase the number of referenced works, as well as the use of a greater number of current works on the subject.
Author Response
- In this sense, it is considered that the manuscript should clarify the methodologies used, describing them in more detail.
Reply: the methodologies used is Described in the manuscript as per reviewr’s comment
- In the results section, a greater description and discussion of the same is necessary, being supported with references to recent studies and works.
Reply: Additional references were added and highlighted.
- In the case of the 3D-QSAR results, activity results are used, which are not previously presented or cited in the manuscript before their presentation in the figure 8.
Reply: Changes have been made according to reviewers comment
- I consider it convenient to increase the number of referenced works, as well as the use of a greater number of current works on the subject.
Reply: Additional references were added and highlighted.
Reply to General comments
- The English language and style were improved as per reviewer’s comment
- Additional references were added as per reviewer’s comment
- The figures are improved as per reviewer’s comment
- Citations are corrected as per reviewer’s comment
- Additional descriptions are included in many sections as per reviewer’s comment.
Round 2
Reviewer 2 Report
Attached

Author Response
Dear Reviewer,
Thank you for your valuable comments and suggestions for the improvement of our manuscript. herewith I have attached the response file for your reference.
